# Burden and clinical impact of 'neglected' transfusion-transmitted infections in Cameroon: A systematic review and meta-analysis

Megane D. Malieuze Nanfah[1,2], Verónica M. Binam Nkot[1,2], Michel M. Yop Kite[1,2], Sandrine S. Beack Bayengue[2], Gertrude Bsadjo Tchamba[3], Alphonse G. Tandja[2], Brice L. Koloko[4], Elodie T. Ngo Malabo[4], Judith G. L. F. Ekwe Priso[2], Elisee L. Embolo Enyegue[5], Martin L. Koanga Mogtomo[1,2], Loick P. Kojom Foko[1,2]*

**1** Center for Expertise and Research in Applied Biology (CEREBA), Douala, Cameroon, **2** Department of Biochemistry, Faculty of Sciences, The University of Douala, Douala, Cameroon, **3** Department of Microbiology and Parasitology, University of Buea, Buea, Cameroon, **4** University Institute of Technology, The University of Douala, Douala, Cameroon, **5** Center for Research on Health and Priority Diseases, Ministry of Scientific Research and Innovation, Yaoundé, Cameroon

\* kojomloick@gmail.com

## Abstract

### Background

Blood supply is a public health challenge in African countries. In Cameroon, blood selection guidelines focus on four viral and bacterial pathogens (HIV, hepatitis B and C viruses, *Treponema pallidum*) associated with transfusion-transmitted infections (TTIs). Other pathogens, often endemic (e.g., *Plasmodium* spp.), are not routinely screened in blood banks and are not included in blood safety guidelines or surveillance, despite their potential for transfusion transmission.

### Materials and Methods

Here, we conducted a systematic review and meta-analysis of prevalence, determinants, and clinical impact of 'neglected' pathogens, defined as pathogens not included in national blood safety guidelines (e.g., filaria, dengue virus, *Toxoplasma gondii*), in blood banks. Additionally, we identified the most urgent challenges and proposed actionable solutions to guide blood safety guidelines in the country.

### Results

A total of 18 studies, covering ~12,500 donations, were included, with the bulk coming from donors living in three regions (Littoral, Northwest, Centre). *Plasmodium* parasite (68.4%) was the major studied pathogen, even though an evident publication bias was found (*p* = 0.004). The other pathogens included dengue virus (5.3%), *T. gondii* (5.3%), and HTLV-1 (5.3%). The filarial parasite *Loa loa* was consistently accidentally found. Even though there is no evidence of SARS-CoV-2-associated TTIs till

**Data availability statement:** All data are within the manuscript and supporting files.

**Funding:** The author(s) received no specific funding for this work.

**Competing interests:** The authors have declared that no competing interests exist.

now, the pooled proportion of this virus was 17.7%. The pooled proportions of infection in blood donors were 16.6% for *Plasmodium* spp. and 0.5% for *Loa loa*. There is a paucity of clinical impact studies on these 'neglected' TTIs, and the available literature suggests impaired levels of immunoglobulin E and albumin. We identified urgent challenges, including awareness among healthcare providers and policymakers, diagnostic and logistical constraints, and low microbial density infections, associated with neglected pathogen-related blood safety.

## Conclusion

We opine that providing more epidemiological evidence is crucial to address the above-mentioned challenges for guiding and guaranteeing blood safety in Cameroon.

### Author summary

Blood transfusion is a vital and life-saving medical procedure during which an individual (recipient) will receive blood products from another individual (donor). Unfortunately, this procedure is associated with transmission risks of infectious pathogens, especially in developing countries, such as Cameroon. While the screening of pathogens such as human immunodeficiency virus (HIV), viral hepatitis, and *Treponema pallidum* (syphilis) is compulsory in blood banks, little attention is given to other, yet highly endemic, pathogens (e.g., malaria parasites). Here, we have conducted a systematic review and meta-analysis to assess the prevalence, patterns, and determinants of these 'neglected' transfusion-transmitted pathogens in Cameroon. The findings indicate that malaria parasites (16.6%) are the predominant pathogens found in donors/blood bags. Other pathogens, such as dengue virus, *Toxoplasma gondii*, and filarial parasites, are also found. These pathogens are also found in co-infections with the above-mentioned major pathogens, especially HIV and viral hepatitis. This study provides baseline data to understand the epidemiology of these 'neglected' pathogens, identifies key challenges associated with their effective control, and proposes solutions to improve blood safety guidelines in Cameroon.

## Introduction

Blood transfusion is a medical intervention during which whole blood or its components (e.g., platelets) are taken from an individual (i.e., the donor) to another individual (i.e., the recipient). An estimated ~118.5 million blood donations, of which 40% were from developed regions, were recorded by the World Health Organization (WHO) [1]. Each year, several million lives are saved through blood transfusions, along with corollaries such as improved health and greater welfare. In practice, the reasons and recipient groups for blood transfusion are diverse between developing and developed areas. While blood transfusion is commonly administered in the

elderly (~60% of all transfusions) in high-income countries, children under five years of age are the primary recipients in low-income countries [2].

Blood safety is crucial to guarantee efficient blood transfusion in recipients. Developing regions in Africa, the Americas, and Asia are striving to improve their blood transfusion practices and guidelines, given the increasing demand for blood [1]. Till now, especially in these regions, blood transfusion is associated with a higher risk of transfusion-transmitted infections (TTIs), which can compromise recipients' health outcomes. Aside from their natural routes of transmission, several pathogens of diverse origins can be transmitted through blood transfusion. These include prions, viruses (e.g., hepatitis C and B viruses – HCV/HBV, human immunodeficiency virus – HIV), bacteria (e.g., *Treponema pallidum*), fungi (e.g., *Candida* spp.), and parasites (e.g., *Plasmodium* spp., *Toxoplasma gondii*) [3,4]. In practice, priority is generally given to HIV, HCV, HBV, and *T. pallidum* in Africa, especially in countries such as Cameroon. The main reason is the higher dangerousness, chronicity, and long-term deleterious impact on individuals.

Little attention is given to other blood-transmitted pathogens, especially to Apicomplexans such as *Plasmodium* spp., and *T. gondii*. Other major life-debilitating pathogens (e.g., *Loa loa*, dengue virus, parvovirus B9, *Candida* spp., *Histoplasma capsulatum*, *Cryptococcus neoformans*) are also routinely overlooked in blood banks from health facilities of several countries (e.g., Cameroon, India), and not integrated in national blood safety policies and surveillance, despite their potential for transfusion transmission [4,5]. In addition, insufficient epidemiological data and a lack of integration into national blood safety policies and surveillance frameworks further contribute to their under-recognition and inadequate monitoring. Some of them, e.g., *Plasmodium* spp., *T. gondii*, are highly endemic in several parts of the globe, including Cameroon. These pathogens can survive for long periods in unscreened blood bags and asymptomatically in infected recipients. *Plasmodium* spp. parasites can provoke clinical malaria, often leading to severe malaria and deaths [6]. Given their clinical impact in the general population, it is also crucial to determine the blood transfusion risk associated with these 'neglected' pathogens in developing countries such as Cameroon.

To date, to the best of the authors' knowledge, there are no systematic reviews and meta-analyses (SRMA) on the epidemiology and clinical impact of these neglected TTIs in Cameroon. This review was designed to address this gap and propose solutions to the current identified blood transfusion-related challenges in Cameroon.

## Materials and methods

### Registration, guidelines, and ethics

The Preferred Reporting Items for Systematic Reviews and Meta-Analyses (PRISMA) 2020 guidelines were followed to write this review (S1 Table) [7]. The review protocol was submitted to PROSPERO for registration (CRD420251148257).

### Search strategy and eligibility

The identification of potentially relevant papers was made through tailored search strategies using databases (PubMed, African Journals Online-AJOL, The Wiley Online Library, ScienceDirect, and ResearchGate), local and international repositories [i.e., local scientific journals (e.g., Health Sciences & Disease), scientific organizations (e.g., Cameroon Academy of Sciences), websites (e.g., WHO African website)], and engines (i.e., Google Scholar). The searches were conducted in English and French from 1st to 30th September 2025. Boolean operators (OR, AND) were combined with search terms to identify relevant papers. For instance, the following search strategy was submitted to PubMed: (("Transfusion-transmitted infection"[Title/Abstract] OR "transfusion transmissible infection"[Title/Abstract] OR "TTI"[Title/Abstract] OR "blood donor"[Title/Abstract] OR "blood donor screening"[Title/Abstract] OR "blood transfusion"[Title/Abstract] OR "blood transfusion-associated infection"[Title/Abstract]) AND ("neglected"[Title/Abstract] OR "burden"[Title/Abstract] OR "clinical impact"[Title/Abstract] OR "prevalence"[Title/Abstract] OR "incidence"[Title/Abstract] OR "magnitude"[Title/Abstract]) AND (Cameroon[Affiliation] OR Cameroon[Title/Abstract] OR "Central Africa"[Mesh] OR "West Africa"[Mesh])). We included all

peer-reviewed studies (cross-sectional, longitudinal, case reports, case series, and cohort) addressing the epidemiology of neglected TTIs (e.g., malaria, toxoplasma, Trypanosoma, filariasis, Brucella, leishmania, Borrelia, cytomegalovirus, Zika virus, *Loa loa*, Dengue virus, Parvovirus B9, *Candida*, *Fusarium*, *Histoplasma capsulatum*, *Aspergillus*, *Cryptococcus neoformans*, or Malassezia) in Cameroon, written in French and/or English, and published between 2006 and 2025. Studies published before 2006 were excluded from the analysis to ensure the relevance and comparability of findings, in terms of diagnostic technologies, blood safety protocols, and malaria control strategies. Even though there is no evidence of blood transfusion of the severe acute respiratory syndrome coronavirus 2 (SARS-CoV-2) via human blood products [8], we have included it in the searches. Its natural history is still elusive, as witnessed by the significant number of papers published on its natural history till now. Thus, it would not be surprising that its ability to elicit TTIs is evidenced in the upcoming years. Conversely, qualitative studies, conference papers, abstracts, preprints, reviews, theses, editorials, letters, and reports were excluded from this analysis.

## Screening strategy and data extraction

Titles and abstracts of papers were evaluated independently by three reviewers (MDMN, MMYK, and VMBN). At this stage, some studies were directly excluded. If eligible, full-texts were retrieved from the databases. For restricted papers, the corresponding author was contacted to provide the full text. The bibliography of relevant papers was checked to identify additional papers. The studies were selected by three authors (MDMN, MMYK, and VMBN), and any discrepancies were resolved through discussion with the supervising author (LPKF). Data of interest were extracted from each paper and entered into an Excel spreadsheet (Microsoft Office, USA).

## Methodological quality

We used the Joanna Briggs Institute (JBI) Critical Appraisal tools to evaluate the quality and bias risk of eligible studies. The tools are designed to appraise studies based on their study design (e.g., cross-sectional, cohort, case series) [9]. Two reviewers evaluated the methodological quality of the included studies.

## Data management

Extracted data from the studies included in the present SRMA were summarized using graphics, tables, narratives, and meta-analysis. All graphics and maps were generated using GraphPad v8.02 and QGIS v3.36.1. Data were extracted and presented at the most granular scale. For instance, if a study used three techniques to detect *Plasmodium* spp. parasites, then, findings on infection prevalence were considered as three independent data points. Regarding data on associated factors, only studies that performed adapted statistical analyses (e.g., logistic regression models, generalized estimating equations) were retained to extract estimators of the magnitude of association (e.g., odds ratios – OR).

## Meta-analysis

We performed a meta-analysis of proportions using the prevalence data of each pathogen. Data extracted to compute pooled estimates were the total number of donors positive for a given pathogen and the total number of screened donors. A minimum of 2 studies was needed to perform meta-analyses, as previously recommended. Forest plots were used to depict pooled estimates using the OpenMeta[Analyst] software v0.24.1 and the online platform MetaAnalysisOnline.com [10]. Meta-analyses were performed using a random-effects model (DerSimonian-Laird approach) with the Hartung-Knapp adjustment to control heterogeneity, and the Freeman-Tukey double arcsine transformation to estimate and stabilize variance. This approach was preferred to logit transformation owing to variance instability when proportions approach 0 or 1, especially when computing pooled estimates for low endemic pathogens. The between-study heterogeneity was assessed and interpreted by computing $I^2$ and performing the Cochran's Q-test [11]. When appropriate, a subgroup and

leave-one-out analyses were used to evaluate the impact of potentially confounding variables (e.g., screening test, donor type) on the pooled estimates. The Egger's test for small-study effects and funnel plots were used to evaluate the publication bias [12]. All statistical analyses were considered significant for *p*-value < 0.05.

## Results

### Characteristics of the included studies

A total of 531 records were identified via database search strategies, of which 186 were excluded. Thus, the full-texts of 345 records were screened for eligibility. Several records were excluded for various reasons (e.g., studies focused on HIV, HCV, HBC, and *T. pallidum*). Fifteen papers were judged eligible for the study, in addition to three papers identified after inspection of bibliographies. We therefore included 18 studies in the systematic review, of which 14 were relevant for meta-analysis (Fig 1A) [13–30]. Most of the studies were cross-sectional (94.4%), conducted in a single study site (94.4%), conducted in the town of Yaoundé (38.9%), and in a clinical setting (88.9%) (Fig 2 & Table 1). Regarding blood donors, 41.2% of studies included family and voluntary donors, and the donor type was not specified in 4 studies (Table 1). *Plasmodium* spp. parasites (68.4%) were mainly targeted in the included studies (Fig 1B), followed by the SARS-CoV-2 (10.5%). It should be noted that the filarial parasite *Loa loa* was accidentally found, i.e., its detection was not included in the predefined study protocols, by the authors of two included studies [26,28]. Peripheral blood film (PBF) (42.3%) and rapid diagnostic tests (RDT) (34.6%) were the main screening techniques (Fig 1C). The bias risk of the studies was good for four criteria related to the objective definition and measurement of exposure and outcome (i.e., clear description of inclusion criteria, valid and reliable diagnostic tools, and utilization of standard criteria to assess neglected TTIs). However, 94.4% of the studies failed to provide site details (i.e., related to blood transfusion), and 66.7% did not use an appropriate statistical analysis to correctly address their defined objectives (S2 Table).

### Pooled overall prevalence of neglected TTIs

Meta-analysis to compute pooled overall proportions was possible for *Plasmodium* spp., *L. loa*, and SARS-CoV-2. The pooled proportion of *Plasmodium* spp. infection in blood donors was 16.6% (95%CI 10.1 – 24.2%, $I^2 = 97.6\%$, $p < 0.001$) (Fig 3A). Publication bias was identified across studies on *Plasmodium* spp. ($p = 0.004$) (S1 Fig). Higher estimates were found for SARS-CoV-2 infection (17.7%, 95%CI 0.0 – 100%, $I^2 = 85.2\%$, $p = 0.009$) (Fig 3B). In contrast, the pooled proportion of *L. loa* infection in blood donors was 0.5% (95%CI 0.1 – 1.1%, $I^2 = 0.0\%$, $p = 0.90$) (Fig 3C). Prevalence data for dengue virus (24.8%), human T-lymphotropic virus 1 (HTLV-1, 5.6%), and *T. gondii* (4.7%) were available for only one study each (Fig 3D).

### Pooled proportion of *Plasmodium* spp. by confounding variables

We evaluated the impact of variables (i.e., period of data collection, towns, region, setting, screening test, and sample size) on the pooled estimates of *Plasmodium* spp. infection. Lower proportions were observed in data collected before 2015 (15.6%, 537/5327, 95%CI 5.7 – 29.2%) compared to data collected between 2016 and 2025 (17.2%, 587/3326, 95%CI 8.3 – 28.4%). The highest pooled estimates were found in the town of Yaoundé (27.6%, 474/1884, 95%CI 11.8 – 47.1%), followed by Buea (23.6%, 367/2848, 95%CI 0.8 – 63.5%), and Douala (14.2%, 177/1314, 95%CI 7.6 – 22.5%). Higher estimates were found using PBF compared to RDT (17.0% *vs* 11.7%, $p < 0.001$) and large-sized studies compared to small-sized studies (17.6% *vs* 16.3%, $p < 0.001$) (S1 Fig).

### Co-infection patterns

Co-infections between major and 'neglected' blood-borne pathogens were reported in four studies [16,21,26,27]. In Douala (Littoral region), Okalla Ebongue *et al*. found 8 co-infections represented by HBV + *P. falciparum* (*n* = 3, 1.2%), *T.*

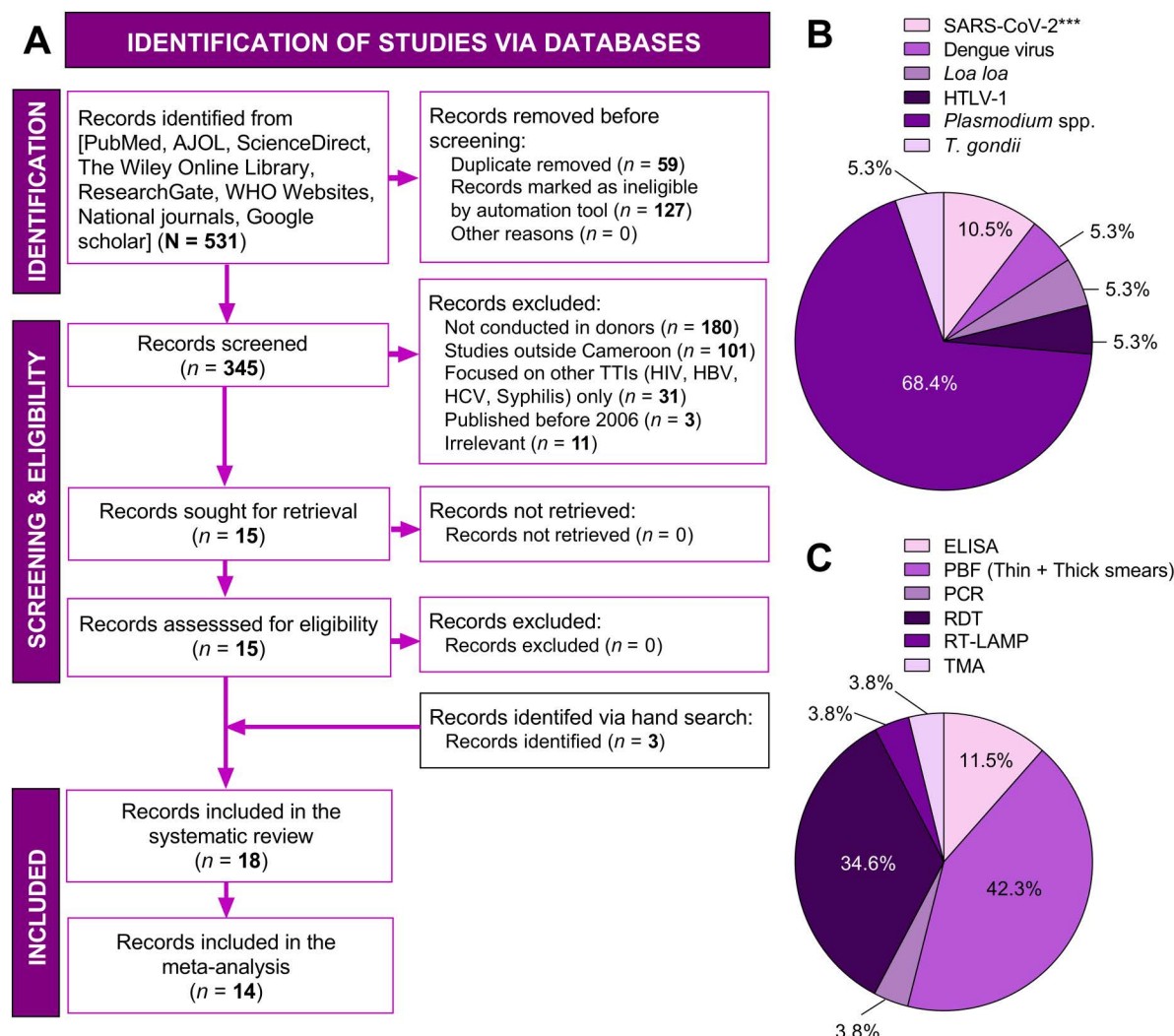

**Fig 1. Study inclusion process as per PRISMA guidelines.** AJOL: African Journals Online, ELISA: Enzyme-Linked Immunosorbent Assay, HIV: Human immunodeficiency virus, HBV: Hepatitis B virus, HCV: Hepatitis C virus, PBF: Peripheral blood film, PCR: Polymerase chain reaction, RDT: Rapid diagnostic test, RT-LAMP: Reverse transcription loop-mediated isothermal amplification, SARS-CoV-2: Severe acute respiratory syndrome coronavirus 2, TMA: Transcription-mediated amplification, WHO: World Health Organization.

*pallidum* + *P. falciparum* ($n = 3$, 1.2%), HCV + *P. falciparum* ($n = 1$, 0.4%), and HIV + *T. pallidum* + *P. falciparum* ($n = 1$, 0.4%) [26]. Focusing on the HTLV-1, Okalla Ebongue *et al*. found 3 co-infections of this virus with HBV ($n = 1$, 0.37%), HIV ($n = 1$, 0.37%), and *T. pallidum* ($n = 1$, 0.37%) in Yaoundé (Centre region) [27]. In Bamenda (Northwest region), two co-infections, i.e., *T. pallidum* + *P. falciparum* ($n = 1$, 0.20%) and HCV + *P. falciparum* ($n = 1$, 0.20%), were reported by Fondoh *et al*. [16], while Tchoffo *et al*. reported the co-infection *T. gondii* + *P. falciparum* at a rate of 0.9% ($n = 3$) [16].

## Residual risk

Only one study evaluated the residual risk associated with TTIs, especially *Plasmodium* spp.-related risk [27]. Okalla Ebongue *et al*. found an individual median residual risk of 5.59 per 10,000 units of blood transfused every year in Douala [27].

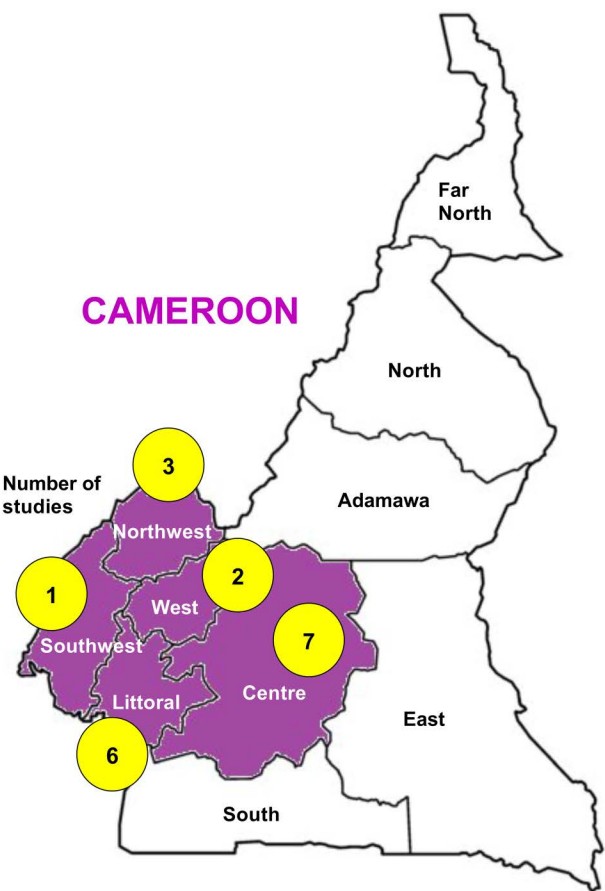

**Fig 2. Map of Cameroon depicting the geographical distribution of the studies.** The Cameroon basemap was downloaded from Natural Earth (Base layer of the map available from: http://www.naturalearthdata.com) and visualized using the QGIS software v3.36.1 (https://qgis.org/en/site/). The regions where the included studies are highlighted in magenta. The total number of studies in each region is shown in the yellow shape. The total number of studies is > 18 because one study was conducted in two regions.

## Associated factors

The associated factors with 'neglected' TTIs (i.e., *Plasmodium* spp.) or putative TTIs (i.e., SARS-CoV-2) were investigated in three studies. In Douala (Littoral region), Medi Sike *et al*. reported higher odds of SARS-CoV-2 infection in donors aged 18 – 28 years (cOR = 3.55, $p < 0.0001$) and 28 – 38 years (cOR = 2.14, $p < 0.0001$) compared to those aged 48 – 58 years [17]. In contrast, Ndoumba *et al*. reported a lower risk of SARS-CoV-2 infection in blood donors aged 35 – 44 years (aOR = 0.44, $p = 0.039$) compared to those aged < 25 years in Bamenda (Northwest region) [18]. In Douala, Ndo *et al*. reported that the risk of *P. falciparum* infection was reduced by ~80% (cOR = 0.71, $p < 0.0001$) among donors sleeping under long-lasting insecticide-treated nets compared with those without them [28].

## Clinical and biological impact of neglected TTIs in donors and recipients

Three studies addressed the clinical impact of TTIs, two on malaria and one on malaria and toxoplasmosis [15,20,21]. In Bafoussam (West region), although not reaching statistical significance, Ewodo *et al*. reported that all *Plasmodium* spp.-infected erythrocyte concentrates failed to meet blood safety and quality standards. Addressing the effect of

**Table 1. Characteristics of studies included in the systematic review and meta-analysis.**

| Year of data collection | Design | Area (Region of Cameroon) | Level of urbanisation | Setting | N | Donor types | Were the study subjects and the setting described in detail? | Was there an appropriate statistical analysis? | Ref. |
|---|---|---|---|---|---|---|---|---|---|
| 1995-2005 | Cross-sectional | Douala (LT) | Urban | Clinical + Community | 1513 | Family + Voluntary | *no* | *no* | [13] |
| 2007 | Cross-sectional | Yaoundé (CEN) | Urban | Clinical | 493 | Family + Voluntary | *no* | *no* | [14] |
| 2014 | Cross-sectional | Douala (LT) | Urban | Clinical | 865 | Voluntary | *no* | *no* | [23] |
| | | Mbingo (NW) | Rural + Suburban + Urban | | 1305 | | | | |
| | | Mutengene (NW) | Rural + Suburban + Urban | | 889 | | | | |
| | | Banso (NW) | Suburban + Urban | | 1170 | | | | |
| 2015 | Cross-sectional | Douala (LT) | Urban | Community | 179 | Voluntary | *no* | *no* | [24] |
| 2015 - 2016 | Cross-sectional | Buea (SW) | Urban | Clinical | 1240 | Not specified | *no* | *no* | [25] |
| 2017 | Cross-sectional | Douala (LT) | Urban | Clinical | 250 | Family + Voluntary | *yes* | *yes* | [26] |
| 2014 | Cross-sectional | Yaoundé (CEN) | Urban | Clinical | 265 | Family + Voluntary | *no* | *no* | [27] |
| 2015 - 2016 | Cross-sectional | Douala (LT) | Urban | Clinical | 372 | Family + Voluntary | *no* | *no* | [28] |
| 2019 | Cross-sectional | Yaoundé (CEN) | Urban | Clinical | 310 | Family + Voluntary + Replacement | *no* | *no* | [29] |
| 2018 | Cross-sectional | Yaoundé (CEN) | Urban | Clinical | 256 | Voluntary | *no* | *yes* | [30] |
| 2021 | Cross-sectional | Bafoussam (W) | Urban | Clinical | 101 | Family + Voluntary + Paid | *no* | *no* | [15] |
| 2020 | Cross-sectional | Bamenda (NW) | Urban | Clinical | 493 | Voluntary | *no* | *not applicable* | [16] |
| 2021 - 2022 | Cross-sectional | Douala (LT) | Urban | Clinical | 102 | Family + Voluntary | *no* | *yes* | [17] |
| 2021 | Cross-sectional | Yaoundé (CEN) | Urban | Clinical | 232 | Family + Voluntary | *no* | *yes* | [18] |
| 2022 | Cross-sectional | Yaoundé (CEN) | Urban | Clinical | 223 | Not specified | *no* | *not applicable* | [19] |
| Not specified | Cross-sectional & Longitudinal | Dschang (W) | Urban | Clinical | 252 | Not specified | *no* | *no* | [20] |
| 2022 | Cross-sectional | Bamenda (NW) | Urban | Clinical | 337 | Not specified | *no* | *no* | [21] |
| 2023 | Cross-sectional | Yaoundé (CEN) | Urban | Clinical | 200 | Family + Voluntary | *no* | *no* | [22] |

CEN: Centre, LT: Littoral, NW: Northwest, SW: Southwest, W: West.

hemoparasites on immunoglobulin E and albumin, Tchoffo *et al*. have noticed higher immunoglobulin E levels ($p = 0.0001$) in *Plasmodium*-infected donors. Likewise, *T. gondii*-infected donors had higher immunoglobulin E levels ($p = 0.026$) and lower albumin levels ($p = 0.037$) [21]. Finally, based on a longitudinal study in the city of Dschang (West region), Djam Chefor *et al*. found that 4 of the 6 recipients who had received *Plasmodium*-infected blood were positive for malaria. Of the 44 recipients included, the post-transfusion malaria incidence was estimated at 9.09% [20].

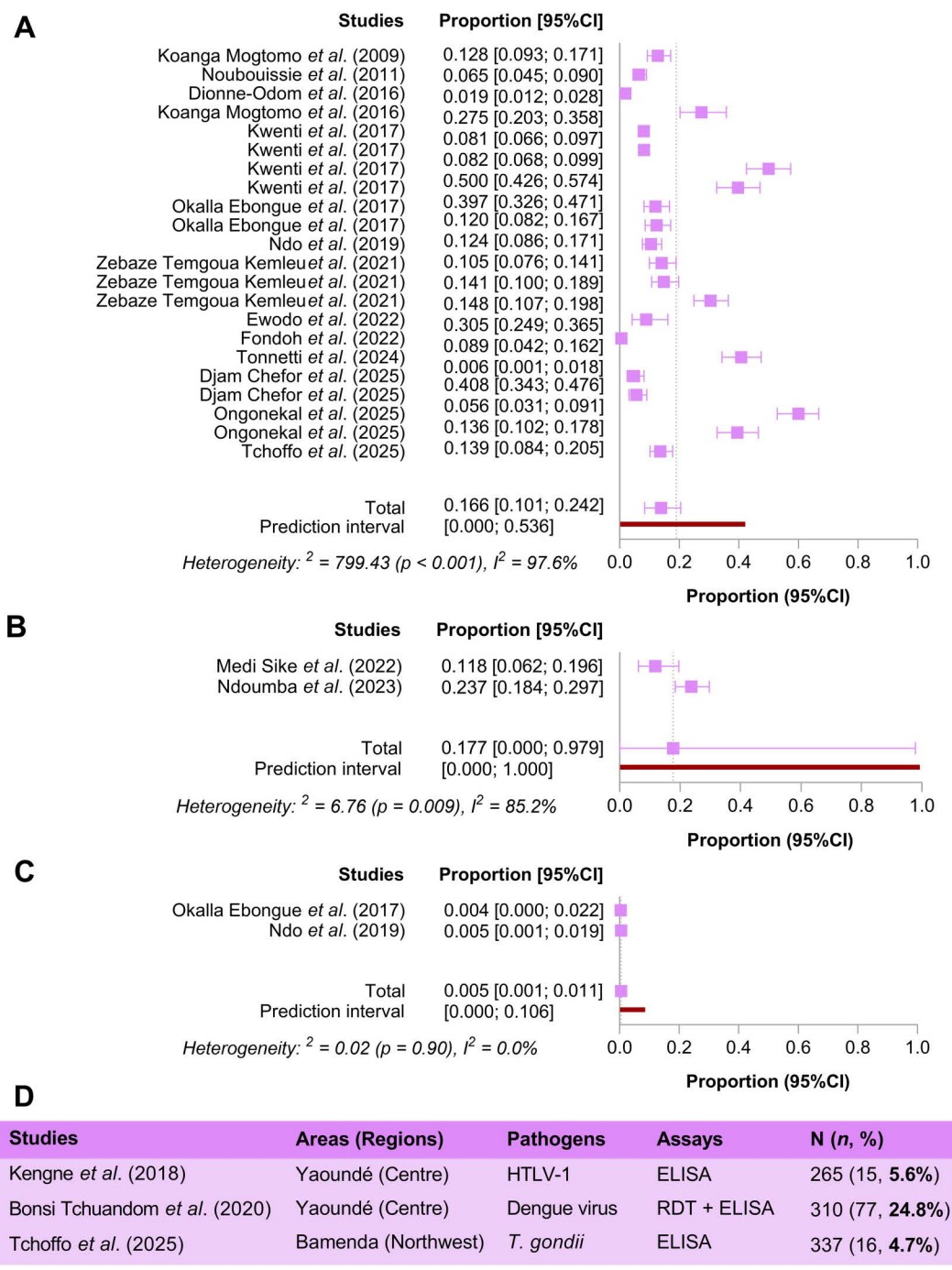

**A**

| Studies | Proportion [95%CI] |
|---|---|
| Koanga Mogtomo *et al.* (2009) | 0.128 [0.093; 0.171] |
| Noubouissie *et al.* (2011) | 0.065 [0.045; 0.090] |
| Dionne-Odom *et al.* (2016) | 0.019 [0.012; 0.028] |
| Koanga Mogtomo *et al.* (2016) | 0.275 [0.203; 0.358] |
| Kwenti *et al.* (2017) | 0.081 [0.066; 0.097] |
| Kwenti *et al.* (2017) | 0.082 [0.068; 0.099] |
| Kwenti *et al.* (2017) | 0.500 [0.426; 0.574] |
| Kwenti *et al.* (2017) | 0.397 [0.326; 0.471] |
| Okalla Ebongue *et al.* (2017) | 0.120 [0.082; 0.167] |
| Okalla Ebongue *et al.* (2017) | 0.124 [0.086; 0.171] |
| Ndo *et al.* (2019) | 0.105 [0.076; 0.141] |
| Zebaze Temgoua Kemleu *et al.* (2021) | 0.141 [0.100; 0.189] |
| Zebaze Temgoua Kemleu *et al.* (2021) | 0.148 [0.107; 0.198] |
| Zebaze Temgoua Kemleu *et al.* (2021) | 0.305 [0.249; 0.365] |
| Ewodo *et al.* (2022) | 0.089 [0.042; 0.162] |
| Fondoh *et al.* (2022) | 0.006 [0.001; 0.018] |
| Tonnetti *et al.* (2024) | 0.408 [0.343; 0.476] |
| Djam Chefor *et al.* (2025) | 0.056 [0.031; 0.091] |
| Djam Chefor *et al.* (2025) | 0.136 [0.102; 0.178] |
| Ongonekal *et al.* (2025) | 0.139 [0.084; 0.205] |
| Ongonekal *et al.* (2025) | |
| Tchoffo *et al.* (2025) | |
| Total | 0.166 [0.101; 0.242] |
| Prediction interval | [0.000; 0.536] |

Heterogeneity: $\chi^2 = 799.43$ ($p < 0.001$), $I^2 = 97.6\%$

**B**

| Studies | Proportion [95%CI] |
|---|---|
| Medi Sike *et al.* (2022) | 0.118 [0.062; 0.196] |
| Ndoumba *et al.* (2023) | 0.237 [0.184; 0.297] |
| Total | 0.177 [0.000; 0.979] |
| Prediction interval | [0.000; 1.000] |

Heterogeneity: $\chi^2 = 6.76$ ($p = 0.009$), $I^2 = 85.2\%$

**C**

| Studies | Proportion [95%CI] |
|---|---|
| Okalla Ebongue *et al.* (2017) | 0.004 [0.000; 0.022] |
| Ndo *et al.* (2019) | 0.005 [0.001; 0.019] |
| Total | 0.005 [0.001; 0.011] |
| Prediction interval | [0.000; 0.106] |

Heterogeneity: $\chi^2 = 0.02$ ($p = 0.90$), $I^2 = 0.0\%$

**D**

| Studies | Areas (Regions) | Pathogens | Assays | N (*n*, %) |
|---|---|---|---|---|
| Kengne *et al.* (2018) | Yaoundé (Centre) | HTLV-1 | ELISA | 265 (15, **5.6%**) |
| Bonsi Tchuandom *et al.* (2020) | Yaoundé (Centre) | Dengue virus | RDT + ELISA | 310 (77, **24.8%**) |
| Tchoffo *et al.* (2025) | Bamenda (Northwest) | *T. gondii* | ELISA | 337 (16, **4.7%**) |

**Fig 3. Pooled proportions of *Plasmodium* spp. (A), SARS-CoV-2 (B), and *Loa loa* (C) in blood donors.** Prevalence of HTLV-1 and Dengue viruses in blood donors **(D)**. 95%CI: Confidence interval at 95%, I²: Heterogeneity, HTLV-1: Human T-lymphotropic virus 1, SARS-CoV-2: Severe acute respiratory syndrome coronavirus 2, ELISA: Enzyme-Linked Immunosorbent Assay, RDT: Rapid diagnostic test. ***SARS-CoV-2 was included in the analysis, even though there is limited evidence of transfusion-transmitted infection cases with this virus via human blood products. However, we kept this virus in the analysis, given the many dark areas of the natural history of this virus.

## Discussion

Blood supply is already a public issue in African countries. This section summarizes the most urgent challenges associated with 'neglected' TTIs in Cameroon.

- The reviewed studies were mainly conducted in three regions (i.e., Centre, Littoral, and Northwest). This regional gap, pinpointed in the present study, is the missing link to understanding the epidemiology of these 'neglected' TTIs, and this warrants the need for further studies in the country, especially in regions with little or no data.

- The lack of data on these 'neglected' TTIs, as well as field evidence, especially for dengue, HTLV-1, or filariasis, is the main challenge. In the present SRMA, we have shown that pooled estimates of *Plasmodium* spp. and *T. gondii* infections can be similar to or exceed those of major TTIs. This assumption is supported by findings of a recent SRMA of major blood-transmitted pathogens in Cameroon, which reported pooled proportions of 2.4%, 8.9%, 2.2%, and 3.6% for HIV, HBV, HCV, and *T. pallidum*, respectively [5]. There are limited or no data on the epidemiology of the above-mentioned non-malarial pathogens (i.e., dengue virus, HTLV-1, filariae). Such epidemiological data are instrumental in evaluating the real danger posed by these pathogens and in guiding blood safety policies in blood transfusion services.

- In individuals qualified for blood donations following screening for major pathogens, these 'neglected' pathogens exist asymptomatically, often at low density malaria infections (LDMIs) [31], can be transmitted to recipients and then elicit a clinical episode. It should be interesting to conduct clinical impact and longitudinal studies in recipients in future to inform healthcare personnel and policymakers. These would be helpful to assess the fraction of *Plasmodium* spp. infections attributed to blood transfusion, as well as their clinical impact.

- It is well known that the infectious risk varies by blood donor types (family, paid, voluntary) for primarily targeted TTIs (e.g., viral hepatitis), with the highest risks commonly reported in family donors. This aspect was recently reported in an earlier investigation in Cameroon [5]. In the present study, it was tricky to address the patterns of the proportion of *Plasmodium* spp. infections by type of donor, and this outlines the need for further studies to address this aspect. The current available data on the association between *Plasmodium* spp. infections and donor type are conflicting. For instance, no significant association was found in Nigeria [32], while in Saudi Arabia, a higher risk of *Plasmodium* spp. infections in family donors compared to voluntary donors [33].

- RDT and PBF are the main techniques used in health facilities in Cameroon. Even though blood bank personnel are aware of malaria risk, *Plasmodium* spp. parasites can still be missed during screening with RDTs due to false negative results for reasons such as LDMIs, non-*P. falciparum* species (e.g., *P. vivax*, *P. ovale* spp.), and the presence of *P. falciparum* parasites with deletions in the histidine-rich protein 2 (*pfhrp2*) gene. The bulk of commercial RDTs are designed to target the histidine-rich protein 2, which is only produced by *P. falciparum* parasites [34]. The emergence and spread of *pfhrp2*-deleted *P. falciparum* parasites is a threat to RDT-based diagnosis, given the ability of these parasites to escape *Pf*HRP2-targeting RDTs, and thus leading to false negative RDT results [35]. Several Cameroonian authors have reported these threats in the territory [36,37]. Additionally, cost and resource constraints associated with PBF are cumbersome, thereby reducing the likelihood of additional testing at blood banks. Personnel at blood banks in Cameroon and other African countries are facing significant work-related pressure. In such settings, priority is often given to mandatory tests for the major TTIs (e.g., HIV, HBV, HCV, and syphilis) in the country. The introduction of additional tests, such as those for *Plasmodium* spp. parasite detection requires more time, resources, and technical capacity. Thus, testing of neglected TTIs is not routinely performed at blood banks of developing settings, such as Cameroon. Besides, point-of-care molecular tests (e.g., Truenat) are available and have shown promising field performance in detecting *P. falciparum* and *P. vivax* LDMIs in other malaria-endemic settings [38]. In this context, it would be interesting to reinforce blood safety via the integration of these point-of-care molecular diagnostic approaches. It is also needed to improve quality assurance systems, enhance staff training, and ensure adequate resource allocation for malaria screening in

blood banks. These call for cost-effectiveness and qualitative studies to evaluate the potential impact of these solutions in the Cameroonian context.

- Many co-infection cases were reported across the reviewed studies, mainly *P. falciparum* with viral infections (i.e., HBV, HCV, HIV). These viral infections, especially HIV, are well known to be a risk factor for malaria severity and deaths [39,40]. Conversely, *Plasmodium* spp. infection is associated with higher HIV viral load and inflammatory responses [39,40]. In a context where persons living with HIV, which could also include pregnant women, can also benefit from blood transfusion, preventing transfusion-transmitted malaria is crucial. In addition, even though limited evidence is available on the impact of anti-retroviral treatment on *P. falciparum*-related antimalarial drug treatment efficacy [41], it is still important to prevent post-transfusion *P. falciparum* malaria in HIV-infected recipients.

- Training of health personnel of blood banks for the screening of these 'neglected' pathogens is essential. Awareness campaigns and periodical training sessions should be implemented and/or scaled up to increase testing rates for these 'neglected' pathogens. A recent study by Nguemnang *et al*. pinpointed very low testing rates for *Plasmodium* spp. and *T. gondii* in blood donors (7% and 4.7%) and recipients (0% and 2.3%) of health facilities in the West region [42].

- Targeting additional pathogens could worsen the blood supply and increase the risk of shortages. There is a balance to find between blood supply and safety. For malaria, it was proposed to give recipients a preventive antimalarial drug. Currently available data indicate a very low rate (9.1%) of prophylactic treatment for post-transfusion malaria in Cameroon health facilities [20]. However, to be more effective, this approach requires access to updated data on drug resistance epidemiology and therapeutic efficacy studies.

### Limitations

The findings of the present SRMA should be interpreted in light of its limitations. First, the limited number of data, most of which came from the Littoral, Northwest, and Centre regions, hindered our ability to generalize our findings to the rest of the regions. Second, we found substantial between-study heterogeneity in meta-analyses. Furthermore, this heterogeneity is likely to arise from differences in study design, population characteristics, geographic coverage, and diagnostic method, and thus, these differences may have influenced the pooled proportion estimates. Indeed, it is well known that the *Plasmodium* spp. transmission is seasonal in several regions of Cameroon [43]. Given the fact that the data collection period varied across the included studies, this may also explain the high between-study heterogeneity observed in the present study. This high heterogeneity is common in meta-analyses of proportions [44]. Third, even though peer-reviewed, we have noticed that several studies were published in poor-quality journals. In general, there is a strong correlation between reviewing rigor and journal quality. Fourth, the few studies that addressed determinants of 'neglected' TTIs have negatively impacted evidence synthesis. Finally, for the handful of studies that have also included recipients, it was not clear if *Plasmodium* spp. infections were due to natural transmission or blood transfusion. Malaria is highly endemic in Cameroon [45]. In such contexts, recipients may acquire infection through mosquito exposure independent of transfusion. Relevant included studies have traced back the origin of *Plasmodium* spp. infections in recipients by performing a pre-transfusion screening, follow-up, and RDT-based testing. As discussed earlier, it is possible that the recipients were already infected with *Plasmodium* spp. parasites before transfusion, with parasites present at densities below the limit of detection of the screening tests. These facts also constitute a limitation of our study.

### Conclusion

This study was designed to conduct a systematic review and meta-analysis of less prioritized blood-borne pathogens in blood blanks of Cameroon. The study revealed a dearth of epidemiological data, with an overrepresentation in three regions of the country. The prevalence of pathogens, such as *Plasmodium* spp. and *T. gondii*, can often outnumber that of

major TTIs. Also, we argue that the most critical challenge is to produce high-quality epidemiological data on the residual risk, burden, extent, and determinants of these 'neglected' transfusion-transmitted pathogens. Disposing of such data is pivotal to addressing the other challenges identified in the review.

## Supporting information

**S1 Table. PRISMA 2020 Checklist.**
(DOCX)

**S2 Table. Bias risk assessment of the included studies.**
(DOCX)

**S1 Fig. Subgroup analysis, sensitivity analysis, and publication bias of studies on malaria in blood donors.**
(DOCX)

## Author contributions

**Conceptualization:** Loick P. Kojom Foko.

**Data curation:** Megane D. Malieuze Nanfah, Verónica M. Binam Nkot, Michel M. Yop Kite, Sandrine S. Beack Bayengue, Gertrude Bsadjo Tchamba, Alphonse G. Tandja, Brice L. Koloko, Elodie T. Ngo Malabo, Judith G.L.F. Ekwe Priso, Elisee L. Embolo Enyegue, Loick P. Kojom Foko.

**Formal analysis:** Megane D. Malieuze Nanfah, Verónica M. Binam Nkot, Michel M. Yop Kite, Loick P. Kojom Foko.

**Investigation:** Megane D. Malieuze Nanfah, Verónica M. Binam Nkot, Michel M. Yop Kite, Sandrine S. Beack Bayengue, Gertrude Bsadjo Tchamba, Brice L. Koloko, Elodie T. Ngo Malabo, Judith G.L.F. Ekwe Priso, Elisee L. Embolo Enyegue, Loick P. Kojom Foko.

**Methodology:** Megane D. Malieuze Nanfah, Verónica M. Binam Nkot, Michel M. Yop Kite, Sandrine S. Beack Bayengue, Judith G.L.F. Ekwe Priso, Elisee L. Embolo Enyegue, Loick P. Kojom Foko.

**Project administration:** Loick P. Kojom Foko.

**Software:** Loick P. Kojom Foko.

**Supervision:** Martin L. Koanga Mogtomo, Loick P. Kojom Foko.

**Validation:** Megane D. Malieuze Nanfah, Verónica M. Binam Nkot, Michel M. Yop Kite, Martin L. Koanga Mogtomo, Loick P. Kojom Foko.

**Visualization:** Megane D. Malieuze Nanfah, Verónica M. Binam Nkot, Michel M. Yop Kite, Sandrine S. Beack Bayengue, Gertrude Bsadjo Tchamba, Alphonse G. Tandja, Brice L. Koloko, Elodie T. Ngo Malabo, Judith G.L.F. Ekwe Priso, Martin L. Koanga Mogtomo, Loick P. Kojom Foko.

**Writing – original draft:** Megane D. Malieuze Nanfah, Verónica M. Binam Nkot, Michel M. Yop Kite, Sandrine S. Beack Bayengue, Brice L. Koloko, Judith G.L.F. Ekwe Priso, Elisee L. Embolo Enyegue, Loick P. Kojom Foko.

**Writing – review & editing:** Sandrine S. Beack Bayengue, Gertrude Bsadjo Tchamba, Alphonse G. Tandja, Brice L. Koloko, Elodie T. Ngo Malabo, Judith G.L.F. Ekwe Priso, Elisee L. Embolo Enyegue, Martin L. Koanga Mogtomo, Loick P. Kojom Foko.

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
