## [Decision Letter · Decision Letter 0]

5 Apr 2026

PNTD-D-26-00009

Burden and clinical impact of ‘neglected’ transfusion-transmitted infections in Cameroon: a systematic review and meta-analysis

Dear Dr. Foko,

Thank you for submitting your manuscript to PLOS Neglected Tropical Diseases. After careful consideration, we feel that it has merit but does not fully meet PLOS Neglected Tropical Diseases's publication criteria as it currently stands. Therefore, we invite you to submit a revised version of the manuscript that addresses the points raised during the review process.

Please submit your revised manuscript within by Jun 04 2026 11:59PM. If you will need more time than this to complete your revisions, please reply to this message or contact the journal office at plosntds@plos.org. Please include the following items when submitting your revised manuscript:

We look forward to receiving your revised manuscript.

Kind regards,

Rana Nagarkatti

Guest Editor

Hira Nakhasi

Section Editor

Shaden Kamhawi

co-Editor-in-Chief

Paul Brindley

co-Editor-in-Chief

**Additional Editor Comments:**

Dear Dr. Foko,

Thank you for submitting your manuscript for review to PLoS NTD. Although you have addressed an important public health issue in Cameroon, the manuscript has significant issues related to description of data, limitations, and analysis of the underlying studies. In addition to the issues raised by the reviewers, please address the following:

1. On line 212-213, you state that “94.4% of the studies failed to provide site details (i.e., related to blood transfusion), and 66.7% did not use an appropriate statistical analysis” with the study details provided in S2 Table.

a. This is a significant issue as it raises questions on the conclusion drawn and the information presented in Table 1. Please move the column on site details and appropriate statistical analysis from S2 Table to Table 1.

b. Please provide appropriate descriptive titles for the tables.

c. The limitations section of your manuscript lacks detail and does not attribute limitations to various factors addressed in S2 Table. The limitations section needs to be re-written.

2. For proportions provided, e.g., line 247 for pooled estimates of Plasmodium infection 15.6%, please provide the numerator and denominator in addition to the percentage and 95% CI.

3. Table 1 shows the donor selection criteria for all donors together (family, voluntary, and paid). The three categories have different risk profiles as blood donors with respect to TTIs. Although there is limited data, does the risk profile vary for these three groups regarding Malaria transmission e.g., paid donors may not have access to Malaria preventive measures, and therefore would be a higher risk? Is it possible to evaluate the three groups separately based on the available literature?

4. Although Plasmodium parasite was the major pathogen based on your inclusion criteria for publications, please provide data on HIV, HCV, and HBV, as it will help readers understand the scope of the issue better when compared to viral TTIs. Viral TTI data may not necessarily be an analysis of excluded publication but from existing literature.

5. Malaria transmission varies by season. Transmission may have also declined due to control programs.

a. Based on the limited number of publications, please address how these two factors impact your study.

b. Additionally, considering that the donors and recipients reside in a Malaria endemic area, you have not addressed the possibility that recipients may have tested positive due to transmission by a mosquito bite rather than by blood transfusion from a false negative tested donor. From the selected literature, please describe how traceback studies were performed to attribute blood donors to recipients testing positive for Plasmodium.

Significant confounding factors exist and may have been addressed in the selected studies, however, these should also be addressed, either in the discussion section or in the limitations section of your manuscript.

6. Please consider providing a map of Camerron along with Table 1 so it is easier for readers to visualize where the studies were performed. What proportion of the country (area and population) is covered by the analysis performed? This is an important factor to analyze considering the impact on public health policy of your manuscript.

Best regards,

Rana Nagarkatti

**Journal Requirements:**

1) Please provide an Author Summary. This should appear in your manuscript between the Abstract (if applicable) and the Introduction, and should be 150-200 words long. The aim should be to make your findings accessible to a wide audience that includes both scientists and non-scientists. Sample summaries can be found on our website under Submission Guidelines:

3) We note that your Data Availability Statement is currently as follows: "Not applicable.". Please confirm at this time whether or not your submission contains all raw data required to replicate the results of your study. Authors must share the “minimal data set” for their submission. PLOS defines the minimal data set to consist of the data required to replicate all study findings reported in the article, as well as related metadata and methods (https://journals.plos.org/plosone/s/data-availability#loc-minimal-data-set-definition).

4) As required by our policy on Data Availability, please ensure your manuscript or supplementary information includes the following:

**Reviewers' Comments:**

Reviewer's Responses to Questions

**Key Review Criteria Required for Acceptance?**

**Methods**

-Are the objectives of the study clearly articulated with a clear testable hypothesis stated?

-Is the study design appropriate to address the stated objectives?

-Is the population clearly described and appropriate for the hypothesis being tested?

-Is the sample size sufficient to ensure adequate power to address the hypothesis being tested?

-Were correct statistical analysis used to support conclusions?

-Are there concerns about ethical or regulatory requirements being met?

Reviewer #1: The objectives of the study were clearly articulated. The study design is appropriate. The authors performed systematic review and meta-analyses on the studies investigating transfusion transmitted infections already published in peer-reviewed journals. The targeted population resides in Cameroon. Authors chose appropriate statistical analyses which supported conclusions of the study. No ethical concerns were noticed.

Reviewer #2: (No Response)

Reviewer #3: Please see Summary and General Comments

**Results**

-Does the analysis presented match the analysis plan?

-Are the results clearly and completely presented?

-Are the figures (Tables, Images) of sufficient quality for clarity?

Reviewer #1: The results are clearly presented with legible figures and tables.

Reviewer #2: 1. Need to provide description of each sub-figure.

Reviewer #3: Please see Summary and General Comments

**Conclusions**

-Are the conclusions supported by the data presented?

-Are the limitations of analysis clearly described?

-Do the authors discuss how these data can be helpful to advance our understanding of the topic under study?

-Is public health relevance addressed?

Reviewer #1: The data supported conclusions except T. gondii. Authors described the limitations of the study.

Reviewer #2: Yes, conclusion supported by the data presented and public health relevance addressed.

Reviewer #3: Please see Summary and General Comments

**Editorial and Data Presentation Modifications?**

Reviewer #1: Introduction

Line 47: Please indicate all four viral and bacterial pathogens

Line 83: Please indicate the duration of blood donations recorded by WHO

Line 101: Please indicate regions where priority is given for 4 pathogens described.

Line 108-109: Please indicate the location of health facilities and provide specifics about countries implementing national blood safety policies and surveillance

Line 115: Please describe in 1-2 sentences on what aspects the mentioned pathogens are being neglected in developing countries.

Material Methods

Line 128: Though authors stated no ethical authorization is required for the current study, please indicate if any ethical considerations were followed in your protocol while handling the data from previous literature.

Line 161: Please indicate whether the three reviewers (mentioned in line 156) and three authors (mentioned in Line 161) are same.

Line 167: Please indicate how many reviewers assessed methodological quality of each study and whether they worked independently.

Line 169: Clarify the sentence structure. Findings from the included studies or SRMA of the included studies? Expand the abbreviation SRMA and indicate that the SRMA mentioned is the current study.

Results

Line 195: Please indicate the dates of data collection for identified records

Line 199: Please clarify whether you identified two or three papers after inspection of bibliographies.

Line 211: Please indicate all four criteria for which bias risk of the studies was good.

Discussion

Line 297: Please indicate the pooled estimates of major TTIs in Cameroon for comparison. Please clarify whether the statement is true for T. gondii infections. You did not provide a pooled estimate for T. gondii infection in Results section.

Line 300: Please indicate the non-malarial pathogens that you are discussing.

Line 301: So far, there is no empirical evidence presented for SARS-CoV-2 causing transfusion-transmitted infections. Therefore, discussing SARS-CoV-2 with relevant to TTI is not necessary. Please remove this sentence.

Line 317: Please elaborate why deletions in the histidine-rich protein 2 genes are important for false negative results in RDTs.

Line 321: Please elaborate how significant work-related pressure faced by personnel at blood banks in Cameroon affect the likelihood of additional testing for malaria in blood donors

Conclusion

Line 356: please clarify how T. gondii prevalence outnumber that of major TTIs.

Supporting information

S1 Table, Introduction, Objectives, item # 4: Please revise as “Introduction, 3rd and 4th paragraph” under the location where item is reported

Figures

Figure 1: Please remove SARS-CoV2 from the data as there is no empirical evidence presented for this pathogen causing transfusion-transmitted infections.

Table S1: Item#26, under location where item is reported, please revise the title to “Competing interests”. Item#27, under location where item is reported, please revise the title to “Data availability statement” and indicate that it is not applicable.

Reviewer #2: (No Response)

Reviewer #3: Please see Summary and General Comments

**Summary and General Comments**

Reviewer #1: The authors performed a systematic review and meta-analysis to assess the burden and clinical impact of ‘neglected’ transfusion-transmitted infections in Cameroon. The authors identified Plasmodium parasite is the major studied pathogen even though an evident publication bias was found. The pooled proportion of Plasmodium spp. infections was 16.6% and substantial between-study heterogeneity was found. The authors indicated paucity of clinical impact studies for Plasmodium and other neglected transfusion-transmitted infections in Cameroon.

Reviewer #2: 1. In 64, you noted that "The pooled proportions of infection in blood donors were 16.6% for Plasmodium spp., 17.7% for SRAS-CoV-2, and 0.5% for Loa loa". Please clarify, if SARS-CoV-2 is transfusion transmitted.

2. Please provide a discussion on what can be done to mitigate the knowledge gap in real-life case vs different data bases.

Reviewer #3: In this manuscript, Malieuze Nanfah et al., provide a review of the burden and clinical impact of some neglected transfusion transmitted infections in the country of Cameroon, Africa. The authors performed a systematic literature review and meta-analysis of prevalence, determinants and clinical impact of pathogens not included in the blood safety guidelines in blood banks in Cameroon, referred as “neglected” in this manuscript (e.g. filaria, dengue virus, Toxoplasma gondii). The authors identified the most urgent challenges and proposed actionable solutions to guide blood safety guidelines in the country.

General comments

1) While this meta-analysis is constrained by a limited number of eligible studies, the scarcity of data is itself a significant finding. Further, as mentioned by the authors, as the available research is restricted to three regions of Cameroon, the results may not be fully generalizable to the national level. These limitations are addressed in the discussion; however, documenting this regional data gap is a primary contribution of this report and warrants publication.

2) Considering the limitations of the studies, the authors identified challenges which should be addressed to improve blood safety in Cameroon. These include awareness of neglected diseases among healthcare providers and policymakers, diagnostic and logistical constraints, and low grade infections.

The authors should address the following comments in their revised manuscript:

3) Line 65: Define the abbreviation “TTIs” in the abstract

4) Line 152: Please explain why publications prior to 2006 were excluded for this analysis.

5) Line 208: Please clarify the meaning of “accidentally found” in the context of the 2 published studies.

6) Lines 254-263: Please discuss the impact of these co-infections on blood safety.

7) Lines 313-324: While this paragraph underscores the operational challenges and limited efficacy associated with RDT and PBF testing, a clear statement regarding the authors' proposed solution is currently missing. Explicitly detailing these improvements would clarify the study's contribution.

8) Line 322: Please briefly explain the “significant work-related pressure” experienced by personnel at blood banks in Cameroon.

PLOS authors have the option to publish the peer review history of their article (what does this mean?). If published, this will include your full peer review and any attached files.

**Do you want your identity to be public for this peer review?** For information about this choice, including consent withdrawal, please see our Privacy Policy.

Reviewer #1: No

Reviewer #2: No

Reviewer #3: No

**Figure resubmission:**
---

## [Decision Letter · Decision Letter 1]

14 May 2026

Dear Dr. Foko,

We are pleased to inform you that your manuscript 'Burden and clinical impact of ‘neglected’ transfusion-transmitted infections in Cameroon: a systematic review and meta-analysis' has been provisionally accepted for publication in PLOS Neglected Tropical Diseases.

Best regards,

Rana Nagarkatti

Guest Editor

Hira Nakhasi

Section Editor

Shaden Kamhawi

co-Editor-in-Chief

Paul Brindley

co-Editor-in-Chief

Reviewer's Responses to Questions

**Key Review Criteria Required for Acceptance?**

**Methods**

-Are the objectives of the study clearly articulated with a clear testable hypothesis stated?

-Is the study design appropriate to address the stated objectives?

-Is the population clearly described and appropriate for the hypothesis being tested?

-Is the sample size sufficient to ensure adequate power to address the hypothesis being tested?

-Were correct statistical analysis used to support conclusions?

-Are there concerns about ethical or regulatory requirements being met?

Reviewer #1: (No Response)

Reviewer #3: The authors have adequately addressed my comments/question in their revised manuscript.

**Results**

-Does the analysis presented match the analysis plan?

-Are the results clearly and completely presented?

-Are the figures (Tables, Images) of sufficient quality for clarity?

Reviewer #1: (No Response)

Reviewer #3: The authors have adequately addressed my comments/question in their revised manuscript.

**Conclusions**

-Are the conclusions supported by the data presented?

-Are the limitations of analysis clearly described?

-Do the authors discuss how these data can be helpful to advance our understanding of the topic under study?

-Is public health relevance addressed?

Reviewer #1: (No Response)

Reviewer #3: The authors have adequately addressed my comments/question in their revised manuscript.

**Editorial and Data Presentation Modifications?**

Reviewer #1: (No Response)

Reviewer #3: The authors have adequately addressed my comments/question in their revised manuscript. I recommend this revised manuscript for publication.

Minor comment: "TTI" should be defined the first time it is used in the abstract

**Summary and General Comments**

Reviewer #1: (No Response)

Reviewer #3: The authors have adequately addressed my comments/question in their revised manuscript.

PLOS authors have the option to publish the peer review history of their article (what does this mean?). If published, this will include your full peer review and any attached files.

Reviewer #1: No

Reviewer #3: No

---

## [Editor Report · Acceptance letter]

Dear Dr Kojom Foko,

We are delighted to inform you that your manuscript, "Burden and clinical impact of ‘neglected’ transfusion-transmitted infections in Cameroon: a systematic review and meta-analysis," has been formally accepted for publication in PLOS Neglected Tropical Diseases.

Best regards,

Shaden Kamhawi

co-Editor-in-Chief

Paul Brindley

co-Editor-in-Chief
